# Membrane Interaction of Ibuprofen with Cholesterol-Containing Lipid Membranes

**DOI:** 10.3390/biom10101384

**Published:** 2020-09-28

**Authors:** Jan Kremkow, Meike Luck, Daniel Huster, Peter Müller, Holger A. Scheidt

**Affiliations:** 1Institute for Medical Physics and Biophysics, Leipzig University, Härtelstr. 16-18, D-04107 Leipzig, Germany; hilbert.da@gmx.de (J.K.); daniel.huster@medizin.uni-leipzig.de (D.H.); 2Department of Biology, Humboldt University Berlin, Invalidenstr. 42, D-10115 Berlin, Germany; luckmeik@cms.hu-berlin.de (M.L.); peter.mueller.3@rz.hu-berlin.de (P.M.)

**Keywords:** ibuprofen, cholesterol, membrane interaction, NMR, fluorescence, membrane properties

## Abstract

Deciphering the membrane interaction of drug molecules is important for improving drug delivery, cellular uptake, and the understanding of side effects of a given drug molecule. For the anti-inflammatory drug ibuprofen, several studies reported contradictory results regarding the impact of ibuprofen on cholesterol-containing lipid membranes. Here, we investigated membrane localization and orientation as well as the influence of ibuprofen on membrane properties in POPC/cholesterol bilayers using solid-state NMR spectroscopy and other biophysical assays. The presence of ibuprofen disturbs the molecular order of phospholipids as shown by alterations of the ^2^H and ^31^P-NMR spectra of the lipids, but does not lead to an increased membrane permeability or changes of the phase state of the bilayer. ^1^H MAS NOESY NMR results demonstrate that ibuprofen adopts a mean position in the upper chain/glycerol region of the POPC membrane, oriented with its polar carbonyl group towards the aqueous phase. This membrane position is only marginally altered in the presence of cholesterol. A previously reported result that ibuprofen is expelled from the membrane interface in cholesterol-containing DMPC bilayers could not be confirmed.

## 1. Introduction

The binding to and molecular interactions of small amphiphilic drug molecules with lipid membranes are of fundamental interest for tailored drug delivery, prevention of unwanted side effects, and alterations of membrane properties which can also influence the biological function of non-cognate receptors [1,2]. For the very popular non-steroidal anti-inflammatory drug ibuprofen (Figure 1), a number of studies have been published describing its membrane partitioning on the basis of the results of various biophysical methods [3,4,5,6,7,8,9,10,11].

One very interesting finding from these studies is a dramatic difference of the membrane position of ibuprofen in cholesterol-free membranes as oppose to cholesterol-containing DMPC bilayers in the gel state, which was described by Alsop et al. [3] using X-ray diffraction techniques. These authors found that ibuprofen is expelled from the DMPC membrane in the presence of cholesterol, while the drug was localized in the hydrophobic chain region of phospholipid bilayers in the absence of cholesterol. In contrast, the molecular dynamics simulations reported by Khajeh and Modarress [5] supported a distribution profile of ibuprofen, which was not significantly influenced by the presence of cholesterol in lipid membranes. It should be noted that the membrane localization of ibuprofen in the absence of cholesterol was confirmed in numerous other studies using different experimental approaches as well as molecular dynamics simulations [4,5,6,7,8]. Since the membrane localization and orientation of ibuprofen is of high interest regarding its influence on membrane permeability and the interaction with other membrane-embedded molecules [12], we have used ^1^H MAS NOESY NMR to take a closer look at these contrary results and to compare the position and orientation of ibuprofen in POPC membranes in the absence and in the presence of cholesterol. As previously shown, the quantitative analysis of the cross-relaxation rates determined from NOESY spectra of lipid samples represents a valuable tool to determine the membrane location and orientation of small molecules [13,14]. POPC as a monounsaturated phospholipid provides in advance to DMPC additional ^1^H-NMR signals in the middle chain region, which can be used for the analysis and represents a more physiological model membrane system. The use of solid-state NMR also implicates the advantage of higher hydration levels of the samples compared to scattering methods and can provide therefore conditions, which are somewhat closer to a native environment.

It is well documented that ibuprofen binds to lipid membranes up to high concentrations [9,10,11]. Several studies describe an influence of the drug on the molecular order within the lipid membrane [6,7,9,15], which can finally induce membrane defects [9,16]. Moreover, ibuprofen may trigger the formation of non-bilayer phases of phospholipid assemblies [3,17]. Therefore, we characterize the influence of ibuprofen on membrane properties by static ^2^H and ^31^P-NMR as well as membrane permeation by fluorescence and electron spin resonance (ESR) assays to assure the state and integrity of the lipid membranes to complete the discussion of the localization of ibuprofen in the membrane.

## 2. Materials and Methods 

### 2.1. Materials

Ibuprofen was purchased form Sigma-Aldrich (Taufkirchen, Germany), the phospholipids 1-palmitoyl-2-oleoyl-*sn*-glycero-3-phosphocholine (POPC), 1-palmitoyl-2-(12-[*N*-(7-nitrobenz-2-oxa- 1,3-diazol-4-yl)amino]dodecanoyl]-*sn*-glycero-3-phosphocholine (NBD-PC) and 1-palmitoyl-*d*_31_-2- oleoyl-*sn*-glycero-3-phosphocholine (POPC-*d*_31_) as well as cholesterol (Chol) were purchased from Avanti Polar Lipids, Inc. (Alabaster, Al, USA). 1-palmitoyl-2-(4-doxylpentanoyl)-*sn*-glycero- 3-phosphocholine (SL-PC) was prepared as described [18]. All other chemicals originated from Sigma-Aldrich (Taufkirchen, Germany) and were used without further purification. For the ESR and fluorescence experiments, stock solutions of ibuprofen in ethanol were prepared.

### 2.2. Preparation of Large Unilamellar Vesicles

Large unilamellar vesicles (LUVs) were prepared by the extrusion method according to previously published procedures [19]. Aliquots of lipids were dissolved in chloroform and dried under vacuum. Lipids were resuspended in a small volume of ethanol (final ethanol concentration was below 1% (*v*/*v*)). Subsequently, HBS (Hepes buffered saline, 150 mM NaCl and 10 mM Hepes, pH 7.4) was added in order to obtain a final lipid concentration of 1 mM containing 0.5 mol% NBD-labeled PC (dithionite assay) and 2.5 mM containing 2 mol% SL-PC (ascorbate assay), respectively. The mixture was vortexed. For preparing LUVs, this suspension was subjected to five freeze–thaw cycles followed by extrusion of the lipid suspension 10 times through 0.1 μm polycarbonate filters at 40 °C (extruder from Lipex Biomembranes Inc., Vancouver, Canada; filters from Costar, Nucleopore, Tübingen, Germany).

### 2.3. NMR Sample Preparation

Ibuprofen and the lipids were mixed in the desired molar ratios in chloroform. After evaporation of the solvent, the samples were redissolved in cyclohexane and lyophilized overnight resulting in ~15 mg overall dry sample for ^1^H-NMR and ~3 mg for ^31^P/^2^H-NMR measurements. After hydration with 40wt% D_2_O or H_2_O for ^1^H-NMR and ^2^H-NMR measurements, respectively, the samples were equilibrated by ten freeze–thaw cycles and gentle centrifugation. In the end, the samples were transferred into 4 mm HR MAS rotors with spherical Kel-F (15 µL volume) inserts for ^1^H-NMR or 5 mm glass vials for ^31^P and ^2^H-NMR or measurements.

### 2.4. Solid-State NMR Measurements

Static ^31^P and ^2^H-NMR spectra were acquired using a Bruker (Bruker Biospin, Rheinstetten, Germany) DRX300 MHz NMR spectrometer using a high-power probe with a 5-mm solenoid sample coil. Proton-decoupled ^31^P-NMR spectra were recorded using a single 3.0 μs 90° pulse, a relaxation delay of 2.5 s, and a sweep width of 500 kHz. To obtain the ^31^P chemical shift anisotropy (Δσ) the resulting spectra were simulated using a program written in Mathcad 2001 (Parametric Technology Corporation, Needham, MA, USA). 

^2^H-NMR spectra were accumulated using the quadrupole echo pulse sequence [20]. The ~3.2 μs 90° pulses were separated by a 50 μs delay; the recycle delay was 1 s, the spectrum width 500 kHz. After depaking the spectra [21], smoothed order parameter profiles where calculated [22].

^1^H-NMR measurements were acquired using a 4 mm double resonance HR-MAS probe with ^2^H lock on a Bruker Avance III 600 MHz NMR spectrometer The MAS frequency was 6 kHz, the π/2 pulse lengths were 4 μs, the relaxation delay was 3 s. The ^1^H-NMR spectra were referenced with respect to the terminal methyl group of the lipid chains at 0.885 ppm for a POPC sample [23]. The 2D ^1^H MAS NOESY spectra were acquired at five mixing times between 0.1 and 500 ms. The volumes of the respective diagonal and cross peaks were integrated using the Bruker Topspin 3.5 software package. Cross-relaxation rates were obtained from fitting the experimental cross peak volumes at varying mixing times using Origin 2017 (OriginLab Cooperation, Northampton, MA, USA) according to the spin-pair-model [14].

All NMR measurements were carried out at a temperature of 30 °C.

### 2.5. Measurement of Dithionite Permeation

The permeation of the anion dithionite across lipid bilayers was measured according to [24,25]. Dithionite quenches the fluorescence of the NBD group of NBD-labeled lipids by a chemical reaction. Upon addition to unilamellar liposomes containing 0.5 mol% NBD-labeled lipids, dithionite rapidly quenches the fluorescence of labeled lipid in the outer membrane leaflet, which is reflected by a rapid initial decrease of fluorescence intensity (see Appendix A). Subsequently, fluorescence decreases slowly caused by a slow permeation of dithionite, reacting with the labeled lipids in the inner membrane leaflet. The assay was performed as described [26,27]. From the reduction kinetics (see Appendix A), the rate constants for the slow decrease (k_p_), i.e., permeation of dithionite were determined. The values of k_p_ in the presence of 0.2 mM ibuprofen (molar ratio lipid/drug = 5:1) were normalized to the respective values in the absence of the drug. The latter measurements, i.e., without drug, were performed with ethanol in the same concentration as used in the presence of ibuprofen (2% *v*/*v*). The experiments were carried out at 37 °C.

### 2.6. Measurement of Ascorbate Permeation

The measurement of ascorbate permeation is similar to the dithionite assay, but here, the reduction of spin-labeled lipids (2 mol%) in liposomes is measured upon addition of ascorbate. The assay was performed as described in [27,28]. From the reduction kinetics, the rate constant k_p_ of the slower component was determined as a parameter for ascorbate permeation. The values of k_p_ in the presence of 1.9 mM ibuprofen (molar ratio lipid/drug = 5:1) were normalized to the respective values in the absence of the drug. The latter measurements, i.e., without drug, were performed with ethanol in the same concentration as used in the presence of ibuprofen (1% *v*/*v*). The experiments were carried out at 37 °C.

### 2.7. Measurement of 6-Carboxyfluorescein Leakage

LUVs were prepared in HBS additionally containing 50 mM 6-carboxyfluorescein [27]. At this large concentration, CF shows a low fluorescence due to self-quenching. CF-filled vesicles were separated from non-incorporated extravesicular CF using PD-10 columns (GE Healthcare, Freiburg, Germany). CF leakage was measured by recording fluorescence of CF in HBS upon addition of ibuprofen at 520 nm over time (λ_ex_ = 490 nm; slit width for excitation and emission, each 4 nm) at 37 °C using an Aminco Bowman Series 2 spectrofluorometer (Urbana, IL, USA). Maximal leakage (I_max_) was determined by the addition of 0.5% (*w*/*v*) Triton X-100. 

## 3. Results and Discussions

To check the phase state of the lipid membrane and to assess the influence of ibuprofen on the phospholipid structure and packing in the bilayer, first stationary ^31^P and ^2^H-NMR measurements were performed. The ^31^P-NMR spectra (see Appendix A) of POPC-d_31_ (in the presence or in the absence of cholesterol, note that phase separation plays not a significant role in POPC/cholesterol systems [29]) exhibit the typical powder pattern line shape of a lamellar bilayer phase in the presence of 10 mol% ibuprofen. Even for higher ibuprofen concentrations (up to 30 mol%) non-lamellar phases were not observed in our preparations (see Appendix A). This is different from previous studies, which have described the formation of non-bilayer phases in different lipid membrane systems at higher drug concentrations. Alsop et al. [3] observed cubic phase formation at an ibuprofen concentration exceeding 10 mol% in dimyristoylphosphatidylcholine (DMPC) membranes using X-ray scattering, and Jaksch et al. [17] detected hexagonal phases above 25 mol% ibuprofen in l-α-phosphatidylcholine membranes using neutron scattering experiments. These differences to our results might not only be attributed to the different lipid systems used, but also be related to the different sample conditions, especially the lower water content required for the application of scattering techniques.

In the presence of ibuprofen, the chemical shift anisotropy (CSA) of the ^31^P-NMR spectra (see Table 1) is significantly reduced compared to the reference samples independent of the presence of cholesterol in the membrane. This decrease indicates an enhanced mobility of the lipid headgroups due to some degree of disordering in this membrane region imposed by the addition of ibuprofen. Further, the orientation of the lipid headgroup relative to the membrane normal could have changed resulting in a similar decrease in the ^31^P chemical shift anisotropy [30,31].

As observed in the ^31^P-NMR spectra, the ^2^H-NMR spectra in the presence of 10 mol% ibuprofen (see Appendix A) also reflect the typical line shape of a lamellar bilayer membrane without any indication for a non-lamellar lipid phase. From these spectra, the lipid chain order parameters for POPC-*d*_31_ were calculated, which show a decrease in the presence of 10 mol% ibuprofen (Figure 2). This decrease, especially in the middle chain region, is even more pronounced in a POPC-*d*_31_/cholesterol (80/20, mol/mol) membrane, where ibuprofen to some extent counteracts the ordering effect of cholesterol. The decreased chain order parameters caused by ibuprofen are also reflected in a decreased lipid chain length (Table 1) calculated according to the analytical models developed in the Brown laboratory [32,34]. These results are in agreement with the larger area per lipid molecule, which has been observed in the presence of ibuprofen by neutron diffraction [7,15,17], Raman microscopy [6], or monolayer experiments [9,33] on different membrane systems. A MD study reported slightly increased order parameters for pure DMPC membranes but a decreased order parameter for DMPC/cholesterol systems in the presence of ibuprofen [5]. Also for other nonsteroidal anti-inflammatory drugs (NSAID), like acetylsalicylic acid or naproxen, comparable influences on the membrane structure were found [6,8,10,35,36].

In spite of the quite drastic effects on lipid chain order, we could show that membrane integrity was not impacted by the addition of ibuprofen. To this end, we used several assays to probe the influence of the drug on the permeation of the polar ions dithionite or ascorbate across POPC and POPC/cholesterol bilayers using fluorescence and ESR spectroscopy (see Appendix A). In both assays, no statistically significant influence of ibuprofen on the permeation rate of the respective molecule could be observed (*p* = 0.05) (Figure 3).

In addition, the leakage of the soluble fluorophore 6-carboxyfluorescein (CF) from the lumen of lipid vesicles upon addition of ibuprofen was measured. This efflux is in principle very slow reflected by a very slow increase of fluorescence intensity. If the bilayer structure is disturbed, e.g., in the presence of membrane-active substances, CF leakage becomes accelerated, which goes along with a faster increase in the detected fluorescence intensity [37]. The curves of time-dependent fluorescence increase for POPC LUV or POPC/cholesterol (80/20, mol/mol) LUVs in the presence of ibuprofen at a molar ratio lipid/drug = 5:1 were very similar to those of the controls, i.e., without the drug (see Appendix A).

While these assays indicate no influence of ibuprofen on the integrity of the membrane, it has to be kept in mind that the sample preparation in the biophysical assays differed from the NMR sample preparation. For the NMR experiments, ibuprofen was incorporated into the lipid membrane through the organic phase before bilayer formation and not added afterwards to the preformed LUVs as done for the permeation/leakage assays. Thus, the question arises, whether this sample preparation procedure could have influenced the results. Therefore, the dithionite permeation experiments were repeated using POPC LUVs prepared by the procedure used for the NMR samples, i.e., by co-solubilization of lipids along with ibuprofen (see Appendix A). Further, these experiments did not reveal any significant difference. Therefore, none of the experiments probing membrane permeation and leakage indicated a loss in membrane integrity induced by the presence of ibuprofen. These results are consistent with a fully intact lamellar bilayer phase.

In order to determine the localization and distribution of ibuprofen in the membrane, ^1^H MAS NMR measurements were performed. Figure 4 displays the 1D ^1^H-NMR spectra of POPC membranes in the absence or in the presence of cholesterol at an ibuprofen concentration of 10 mol%. Besides the well-resolved POPC resonances, also signals arising from ibuprofen are visible. The signals at 7.0 and 7.2 ppm can be assigned to the protons of the aromatic ring of ibuprofen, the signal at 1.8 ppm represents the single proton of the isobutyl group of ibuprofen (Figure 1) [38]. Unfortunately, other ^1^H-NMR signals of ibuprofen overlap with POPC peaks as the lipid β or the signal of the terminal methyl groups of the lipid chains.

A first view on the membrane location of ibuprofen can be obtained from analyzing the induced chemical shifts of the individual POPC ^1^H-NMR signals. Since these alterations in the NMR peak positions depend on the distance between the aromatic ring of ibuprofen and the individual molecular segments of the phospholipid, a plot of the induced chemical shift as a function of the POPC segments along the long axis of the molecule provides an estimation of the average location of the aromatic ring in the membrane [13,39,40]. In the presence of ibuprofen, all lipid signals exhibit an induced chemical shift with the largest values for the C2 and G-1 resonance (Figure 5), pointing to an average location of the aromatic ring in the upper chain glycerol region of the membrane. This position is similar in a cholesterol-containing and a pure POPC membrane (Figure 4).

To obtain a more quantitative insight into the localization and orientation of ibuprofen within the lipid membrane, 2D NOESY spectra with varying mixing times were acquired. These spectra exhibit resolved cross peaks between all lipid signals and both signals of the aromatic ring of ibuprofen (see Appendix A). For the signal at 1.8 ppm, assigned to the single proton of the isobutyl group of ibuprofen, solely cross peaks to the lipid chain region of POPC could be observed. For some of the other lipid signals, very weak and hardly visible NMR signals were detected, preventing further quantitative analysis.

Using a quantitative analysis of the cross peak intensities applying the spin-pair-model [14,41], the cross-relaxation rates between the protons of ibuprofen and the respective molecular segments of POPC were obtained. As it was established for many comparable small molecules [13,14,26,40,42,43], the plot of these cross-relaxation rates as a function of the lipid segments relative to the long axis of the molecule yields to a distribution function of ibuprofen in the lipid membrane (Figure 6). Please note that those POPC signals substantially overlapping with ibuprofen signals, i.e., the terminal CH_3_-group of the acyl chains and the headgroup C-β were excluded from the analysis since an intramolecular contribution to the respective cross peaks cannot be ruled out. All obtained cross-relaxation rate plots (Figure 6) exhibit a broad distribution of the respective molecular groups of ibuprofen within the membrane. This indicates the high molecular disorder as well as pronounced dynamics of the drug molecule in the lipid membrane in the physiologically relevant liquid crystalline state [41,44,45]. The distribution functions of all aromatic protons of ibuprofen exhibit their maxima in the upper chain region (C-2), whereas the single proton of the isobutyl group is located somewhat deeper in the middle chain region of the membrane. The observed orientation of ibuprofen with its nonpolar part (see Figure 1 and Figure 7) deeper in the membrane is a result of the complex pattern of physical interactions between the drug and the phospholipid molecules, which are quite different in the lipid-water interface and in the hydrophobic core of the membrane. The above-mentioned localization is in agreement with previous but less detailed NOESY NMR results in pure POPC membranes [4] and also with data derived from X-ray diffraction [3] or Raman microscopy [6] in pure DMPC as well as MD simulations [5,7]. A similar membrane position was also found for other NSAIDs due to some similarity in their molecular structure leading to comparable physical interactions in the lipid membrane [6,8].

Finally, to decipher potential cholesterol-associated differences regarding the membrane localization of ibuprofen the measurements were also performed for a POPC membrane containing 20 mol% cholesterol. The obtained distribution functions of ibuprofen in the presence of cholesterol show a very small shift of their maxima in the direction of the lipid head group as the only difference in comparison to the data of pure POPC (Figure 6), which is most pronounced for the single proton of the isobutyl group of ibuprofen (1.8 ppm) in the hydrophobic part of ibuprofen. These minimal changes are not consistent with the observation of Alsop et al. [3] using X-ray diffraction, where the authors proposed the expulsion of ibuprofen from the DMPC membrane in the presence of 20 mol% cholesterol. These different observations can be explained by the different membrane compositions used in both studies. While Alsop et al. used the saturated DMPC in the gel state with a very low hydration level, in the current study the membranes consisted of the physiologically more relevant monounsaturated POPC in the liquid crystalline phase state. Of course, both model membrane systems cannot really mimic the high complexity of natural biological membranes and have their advantages and disadvantages depending on the applied methods. However, both lipids used in this study, i.e., POPC and cholesterol, belong to the main components of biological membranes. Additionally, it should be noted that the very low water content of the lipid samples needed for scattering method has a significant influence to the phase state of the membrane and therefore to the molecular interactions. From our data, we hypothesize, that the membrane interaction between lipids and ibuprofen is influenced by the physico-chemical state of the membrane, i.e., in a more rigid and ordered gel state, an insertion of ibuprofen into the membrane is impaired. It should be noted that NOESY measurements of membranes in a gel state are not possible due to the lack of resolution in the ^1^H-NMR spectra. Moreover, in the study of Alsop et al., a lower concentration of ibuprofen (2 mol%) was used compared to the current experiments (10 mol%), which were necessary in order to obtain sufficient NMR signals.

## 4. Conclusions

The current study shows that ibuprofen disturbs the molecular order of liquid crystalline phospholipid membranes by increasing the molecular mobility of the head groups as well as of the acyl chains of the phospholipids. The observed ibuprofen-mediated increase of mobility and disorder in a phospholipid bilayer may promote a further incorporation of the drug into the membrane. The membrane integrity and the lamellar bilayer phase remains intact in the presence of ibuprofen molecules. Membrane-embedded ibuprofen adopts a mean position in the upper chain/glycerol region of the membrane, oriented with its polar carbonyl group towards the membrane surface (Figure 7). The presence of cholesterol in membranes only has a very small effect on the localization of ibuprofen in the model membrane.

## Figures and Tables

**Figure 1 biomolecules-10-01384-f001:**
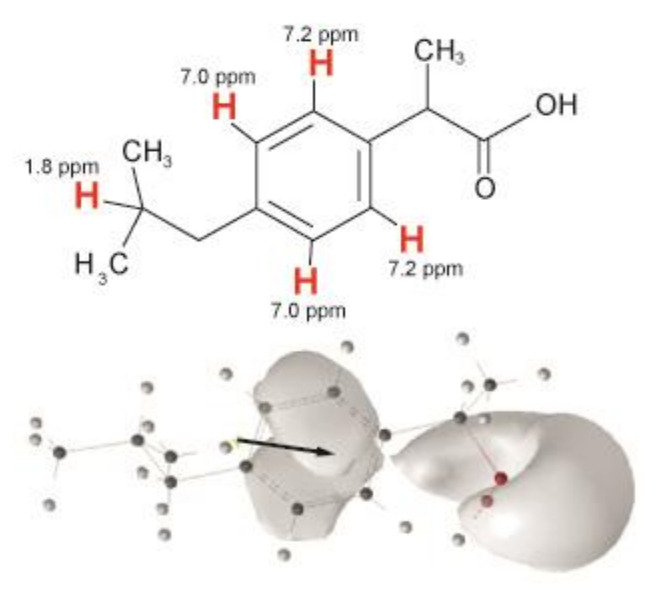
Chemical structure of ibuprofen (**top**) with the assignment of the resolved proton signals (red) and orientation of the dipole moment (black arrow) and isosurface of the electrostatic potential (**bottom**).

**Figure 2 biomolecules-10-01384-f002:**
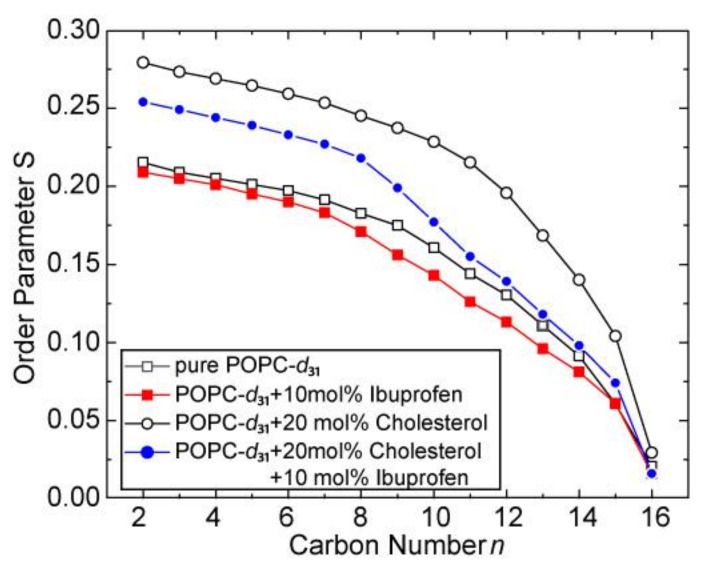
^2^H-NMR chain order parameters plots of POPC-*d*_31_ in the absence and in the presence of 10 mol% ibuprofen and/or 20 mol% cholesterol.

**Figure 3 biomolecules-10-01384-f003:**
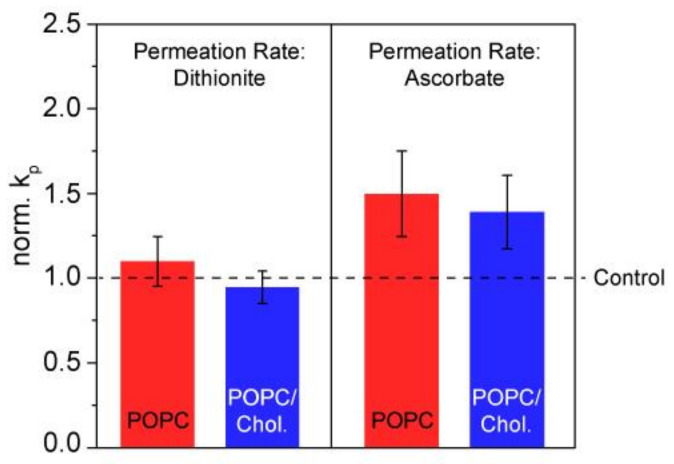
Normalized rate constants (k_P_) for dithionite (left) and ascorbate (right) reflecting the permeation of the respective anion across the membrane in POPC and POPC/Chol (80/20, mol/mol) LUVs. k_P_ values determined in the presence of ibuprofen (molar ratio lipid/drug = 5:1) were normalized to those determined in the absence of the drugs. For the dithionite assay, vesicles contained 1 mM lipid and 0.5 mol% NBD-PC. For the ascorbate assay, the lipid concentration was 2.5 mM and 2 mol% SL-PC. The data represent the mean ± SE of ≥ 6 (dithionite) and 8 (ascorbate) independent experiments.

**Figure 4 biomolecules-10-01384-f004:**
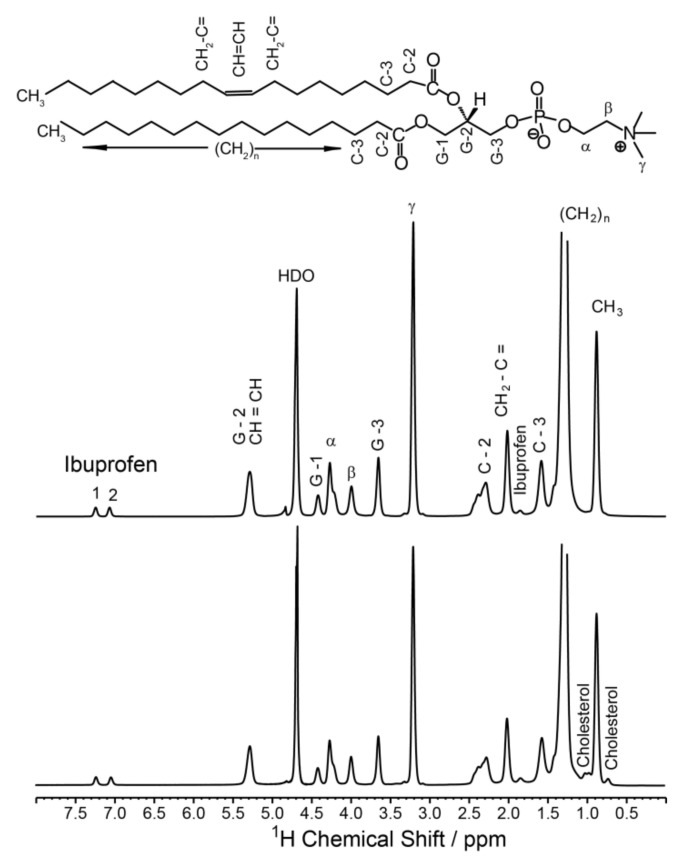
^1^H MAS NMR spectra of POPC membranes (**top**) and POPC/cholesterol (80/20, mol/mol) (**bottom**) in the presence of 10 mol% ibuprofen. Resolved signals of ibuprofen and cholesterol are marked; the assignment of the POPC peaks (nomenclature see chemical structure of POPC above the spectra) was taken from [23]. HDO denotes the signal of residual water. The NMR spectra were measured at a MAS frequency of 6,000 Hz and a temperature of 30 °C.

**Figure 5 biomolecules-10-01384-f005:**
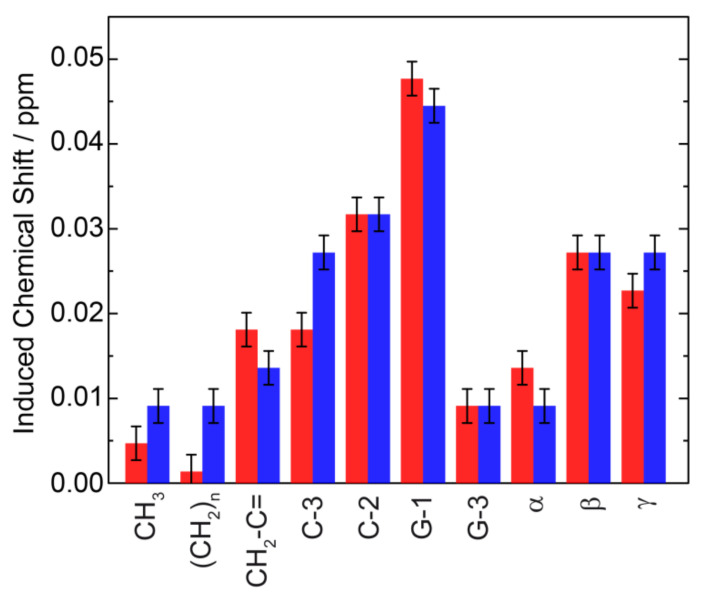
Induced chemical shifts for the ^1^H signals of POPC due to ring current effects in the presence of 10 mol% ibuprofen in a POPC (red) and in a POPC/cholesterol (80/20, mol/mol) membrane (blue).

**Figure 6 biomolecules-10-01384-f006:**
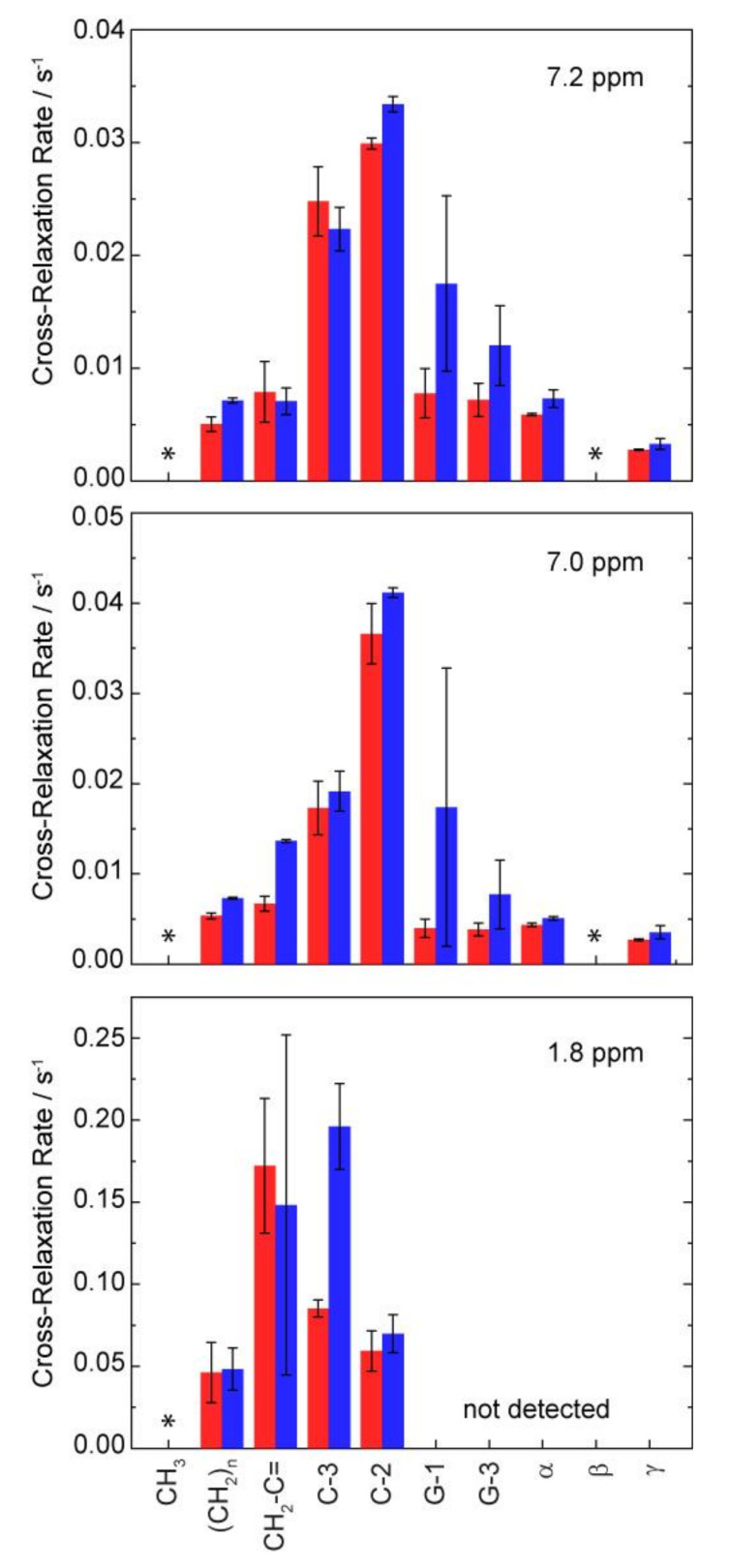
NOESY cross-relaxation rates between the three resolved signals of ibuprofen and the respective molecular segments of POPC along the membrane normal in POPC (red) and in POPC/cholesterol (80/20, mol/mol) (blue) membranes in the presence of 10 mol% ibuprofen. Cross-relaxation rates for the positions marked by * were not analyzed because of a significant signal overlap of ibuprofen and POPC signals. For the ibuprofen signal at 1.8 ppm, only cross peaks for the chain region of POPC were clearly observed.

**Figure 7 biomolecules-10-01384-f007:**
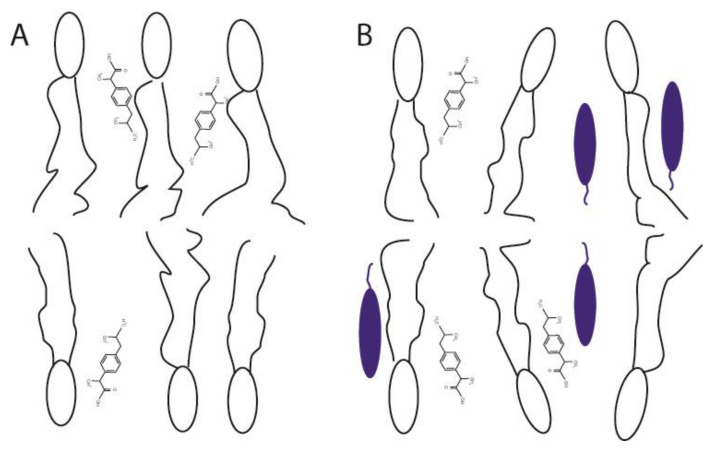
Sketch of the membrane position and orientation of ibuprofen in (**A**) a POPC membrane and (**B**) POPC/cholesterol membrane deduced from the NOESY data in this study. Cholesterol molecules are represented by the blue ellipses.

**Table 1 biomolecules-10-01384-t001:** Chemical shift anisotropy (CSA) Δ*σ* in the ^31^P-NMR spectra and calculated chain extent *L*_c_* [32,33] of POPC-*d*_31_ in the different investigated samples.

Sample	^31^P CSA Δ*σ*/ppm	*L*_c_*/Å
POPC-*d*_31_	45.5	11.7
10 mol% Ibuprofen/POPC-*d*_31_	40.2	11.1
20 mol% Ibuprofen/POPC-*d*_31_	40.2	11.2
30 mol% Ibuprofen/POPC-*d*_31_	38.0	10.4
20 mol% cholesterol/POPC-*d*_31_	44.0	13.2
10 mol% Ibuprofen/20 mol% cholesterol/POPC-*d*_31_	40.2	12.1

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
