# Peer review of "Membrane Interaction of Ibuprofen with Cholesterol-Containing Lipid Membranes"

_biomolecules, 2020, doi:10.3390/biom10101384_

Round 1
Reviewer 1 Report
Dear authors,
Please find below the review report.
Review report
Manuscript ID: 913074
Title: Membrane interaction of ibuprofen with cholesterol-containing lipid membranes
Brief summary:
The manuscript aims at characterizing the influence of ibuprofen on the properties of lipid membranes. Fluorescence and electron spin resonance experiments were performed to evaluate the integrity of the lipid membranes upon ibuprofen incorporation, while solid-state NMR studies were performed to evaluate the structure of the lipid membranes and the location of ibuprofen within the membranes. The studies were performed using POPC and POPC:Cholesterol (8:2) as membrane model systems. Ibuprofen was found to decrease the molecular order of the phospholipids and to be located in the upper chain/glycerol region of the membrane.
Broad comments:
The manuscript meets the scope of the journal as it intends to describe the interactions of a small molecule (ibuprofen) with relevant biomolecules that compose the biological membranes. Overall, the manuscript is well written, even though I believe the materials and methods sections can be improved (as indicated in the specific comments) to facilitate the task of the readers. In order to clarify the starting point of the manuscript in terms of comparison with the existing literature, I would state from the beginning (introduction section) that the lipid membrane used by Alsop et al.
- was made of DMPC in the gel phase, to avoid misunderstandings and to further support the discussion section. In another positive note to improve the discussion section, I believe that the authors can enrich the discussion by comparing the effects observed for ibuprofen with those reported for other NSAIDs, as many other drugs of this class were previously reported, for instance, to have a similar membrane location than that reported for ibuprofen in this
Specific comments:
Please find below some points to improve the clarity of the manuscript:
- Lines 33-35: To improve the clarity for readers I would insert the references of the studies of ibuprofen-membrane interactions at the end of the sentence “For the very popular non- steroidal anti-inflammatory drug ibuprofen (Fig. 1), a number of studies have been published describing its membrane partitioning on the basis of the results of various biophysical ”
- Line 56: I suggest the authors to define ESR at its first appearance in the
- Section 2.1: please clarify the quantity of ethanol used in ESR and fluorescence experiments. Did the authors check that ethanol did not alter the properties of the lipid bilayers?
- Section 5: Please clarify in the material and methods section the quantity of NBD-labeled and spin-labeled lipids used in the measurements of dithionite permeation and ascorbate permeation. Do the properties of the lipid bilayer remain the same with the quantity of labeled lipids used?
- I suggest that section “Measurement of ascorbate permeation” should be formatted as section 2.5 and 6.
- Section 2.6: The ionic strength of 50 mM of 6-carboxyfluorescein in HBS is quite superior to that of When separating the CF-filled vesicles the authors have taken into account the differences of ionic strength of the inner and outer media? These differences may completely hamper the acquisition of proper data.
- Please justify the choice of POPC and cholesterol as model lipids, as well as the concentrations of cholesterol and ibuprofen to be
- Figure 3: according to the data presented in Figure 3, it seems that the permeation rate of ascorbate increases in the presence of ibuprofen in comparison with the control value. Even though, the figure does not show the SD corresponding to control, making the interpretation of results difficult. Can you please clarify?
- Lines 270-272: According to the authors: “While Alsop et al. used the saturated DMPC in the gel state with a very low hydration level, in the current study; the membranes consisted of the physiologically more relevant monounsaturated POPC in the liquid crystalline phase state.” Putatively, POPC bilayers may be more relevant than DMPC bilayers as membrane models. However, both systems are models with advantages and disadvantages. In fact, we should remember the complexity of the composition of biological membranes, as well
as their heterogeneity with various microdomains. In my point of view, both studies give important information for predicting the real effect of ibuprofen in biological systems, considering the systems used and the closer situations being mimicked. In this sense, I believe the authors may improve this section of the discussion by taking in consideration the overall complexity of the biological membranes.
All the best.
Author Response
Please see attachement

Reviewer 2 Report
The authors present NMR and permeation studies of ibuprofen interactions with POPC membranes with and without cholesterol. The main findings involve a location for ibuprofen near the lipid glycerol groups, modest perturbations of lipid chain order, and minor dependence on the presence of cholesterol. The results are important and timely.
The comparisons with results from previous studies should be better and more objectively explained. Several other minor changes would improve this paper.
With regard to differences in observations of non-bilayer phases, the different lipids could be a factor. Because the lipids are different, the NMR results do not strictly contradict previous studies.
In Methods, the total lipid and ibuprofen concentration ranges and sample volumes should be stated for the vesicle and NMR samples.
The yellow arrow does not show in figure 1 of my version of the manuscript file.
The statement “which contradicts the finding of Alsop” (line 236) should be explained. Is this a large or a small contradiction? Were the same lipids represented? The issue is better addressed later in lines 265-278, where the gel state is mentioned. Perhaps the lipid phase is crucial, such that the methods measure different samples and do not really “contradict” ? More generally, a theme of “contradicting” previous measurements seems to be overstated. The sample differences should be emphasized more prominently, for example in the abstract.
“Liquid crystalline phospholipid membranes” should be stated in the first paragraph of the conclusion.
Round 2
Reviewer 1 Report
Dear authors,
I hope you find my previous comments useful to improve your manuscript, which I believe is ready to be accepted in the present form.
All the best.
Author Response
We thank you for your positive evaluation of our work and the constructive remarks.